# Non-Targeted Metabolic Profiling of Cerebellum in Spina Bifida Fetal Rats

**DOI:** 10.3390/metabo13050670

**Published:** 2023-05-19

**Authors:** Evan Thielen, Marc Oria, Miki Watanabe-Chailland, Kristin Lampe, Lindsey Romick-Rosendale, Jose L. Peiro

**Affiliations:** 1The Center for Fetal and Placental Research, Division of Pediatric General and Thoracic Surgery, Cincinnati Children’s Hospital Medical Center (CCHMC), Cincinnati, OH 45229, USA; thieleep@mail.uc.edu (E.T.); marc.oria@cchmc.org (M.O.); kristin.lampe@cchmc.org (K.L.); 2Department of Surgery, College of Medicine, University of Cincinnati, Cincinnati, OH 45267, USA; 3NMR-Based Metabolomics Core, Division of Pathology and Laboratory Medicine, Cincinnati Children’s Hospital Medical Center, Cincinnati, OH 45229, USA; miki.watanabe@cchmc.org (M.W.-C.); lindsey.romick-rosendale@cchmc.org (L.R.-R.)

**Keywords:** metabolism, Spina bifida, myelomeningocele, chiari type II, cerebellum, fetal rat, energy failure, oxidative stress, hypoxanthine, inosine, niacinamide, glutathione, glutamate, succinate, glutamine, phenylalanine, pantothenate, sn-glycero-3-phosphocholine

## Abstract

Spina bifida, known more commonly as myelomeningocele, is a neural tube defect that results in herniation of the cerebellum through the foramen magnum into the central canal as part of the Chiari II malformation. Effects stemming from the herniated cerebellum and its metabolic profile have not been extensively studied. The objective of this study is to examine the metabolic effects of this disease on the cerebellum in utero through the utilization of a retinoid acid-induced Spina bifida rat model. Analysis of this model at mid-late (day 15) and term (day 20) of gestation in comparison to both non-exposed and retinoic acid-exposed non-myelomeningocele controls, the observed metabolic changes suggest that mechanisms of oxidative stress and energy depletion are at play in this neuro tissue. These notable mechanisms are likely to result in further damage to neural tissue as the fetus grows and the compressed cerebellum develops and herniates more due to myelomeningocele.

## 1. Introduction

Spina bifida is classified as a neural tube defect in which the spine and spinal cord are unable to engage in proper closure during the fourth week of embryonic progression. Myelomeningocele (MMC) is the most severe form of Spina bifida, appearing with an open spinal canal that externally exposes tissues and nerves as the development of the fetus progresses, pushing these vital components through the newly formed opening. As a result of chronic cerebrospinal fluid (CSF) leak, the posterior fossa collapses, and the cerebellum can herniate into the central canal extending below the foramen magnum, leading to potential secondary obstruction of the cerebrospinal fluid flow and increased intracranial pressure. This leads to severe consequences, including paralysis, brainstem issues, and hydrocephalus, besides lower extremity malfunction and changes in bladder and bowel function [1]. The Chiari Type II malformation is a unique hindbrain malformation that is nearly omnipresent in all open neural tube defects. The Chiari Type II malformation results in caudal displacement and herniation of the posterior fossa, causing the cerebellar vermis to become increasingly low in displacement and the cerebellum to appear small and thin [2,3]. This malformation has been studied in regard to its impact on hydrocephalus and spinal cord injury, showing a potential role in the development of hydrocephalus through compression of the cerebral aqueduct [4]. However, the effect of Chiari Type II malformation on the herniated cerebellum itself has not been extensively studied. The herniated cerebellum has been looked at using histopathology and various imaging modalities, but the pathophysiology of this malformation’s effect on cerebellar tissue has not been broadly explored at a metabolic level [5,6].

Using a well-established congenital retinoic acid-induced Spina bifida rat model permitted exploring Spina bifida and its resultant Chiari Type II malformation [7,8]. In summary, by gavaging pregnant rats once with retinoic acid 10 days into their 21-day gestational period, as has been reported before, Spina bifida was seen in about 70% of the fetuses [7]. Within this model, a hindbrain malformation that resembles Chiari Type II can be seen and can be utilized for further study of the mechanisms of injury that result as a result of this disease.

The metabolomic analysis serves as a useful quantitative profiling technique that assists in identifying alterations in metabolites that occur throughout the body and in a specific tissue. This analysis is useful in understanding alterations that may be associated with certain mechanisms of injury. Metabolomics has been used in analyzing various aspects of Spina bifida across different animal models, including fetal sheep and mice [9,10]. This technique allows for further investigation into the cause-and-effect relationship between Spina bifida and the pathological changes that may occur.

The objective of this study is to identify the metabolic changes seen in the cerebellar tissue of this Chiari-II -like malformation found in a retinoic acid-induced Spina bifida rat model by nuclear magnetic resonance (NMR) based metabolomics, pathway analysis, and mathematical evaluation of energy data. These metabolic changes can then be utilized to better understand the potential mechanism of injury that are at play in this Chiari Type II malformation that leads to further dysfunction in the cerebellum during the development of myelomeningocele.

## 2. Experimental Design

The following protocols agreed with the National Institutes of Health Guidelines for Care and Use of Laboratory Animals. They were approved by the Institutional Animal Care and Use Committee at Cincinnati Children’s Hospital Medical Center (IACUC #2019-0081).

### Retinoic Acid-Induced Spina Bifida Rat Model

A group of 36 timed-pregnant Sprague–Dawley rats with weights of 200–250 g (Charles River Laboratories, Inc., Wilmington, MA, USA) were housed individually at a temperature of 22 °C with a standard dark:light schedule (12:12 h, light 7:00–19:00) with full access to water and standard food. The Spina-bifida-induced model utilized time-pregnant rats, and the date of mating was labeled as E-1 and the date of plug presence as E0. Trans retinoic acid (RA) (Sigma-Aldrich Chemical, St. Louis, MO, USA) was solubilized at room temperature in olive oil and protected from light, with use taking place within 1 h of its formulation. On gestation day 10 (E10) at 10 am, 20 timed-pregnant rats were fed once with RA 100 mg/kg in olive oil in an attempt to induce Spina bifida in the litter, and 16 timed-pregnant rats were given normal olive oil as a vehicle control [7]. Using this well-established and cost-efficient teratogenic animal model (Supplemental Appendix A) and the matched internal controls (RA) and vehicle (VC) controls allowed us to identify the changes during the herniation of the ecerebellum in the central canal pathophysiology and not due to genetic variants [8].

## 3. Procedure

### 3.1. Tissue Collection

Litters from these timed-pregnant rats were harvested at two time points in gestation, mid-late gestation (labeled E15), and term gestation (E20). Cerebellar tissue samples were collected and classified into one of three groups: Spina bifida (labeled MMC); retinoic-acid sham (RA); or normal (VC). A total of 30 cerebellum samples were collected, with 5 biological replicates from different mothers for each group at each point in gestation (mid-late and term gestation). Cerebellar tissues were snap-frozen and stored at −80 °C. (Appendix A).

### 3.2. Sample Preparation and NMR Data Collection

The modified Bligh and Dyer extraction was used to obtain polar metabolites as described previously [11,12]. The polar phase was dried and resuspended in 220 μ of NMR buffer containing 100 mM phosphate buffer, pH7.3, 1 mM TMSP (3-Trimethylsilyl 2,2,3,3-d4 propionate), 1 mg/mL sodium azide) prepared in D_2_O. The final volume of 200 μ of each sample was transferred into a 3 mm NMR tube (Bruker) for data collection.

All the NMR spectra were acquired on a Bruker Avance III HD 600 MHz spectrometer with a 5 mm BBO Prodigy probe and processed with Topspin 3.6 software (Bruker Analytik, Rheinstetten, Germany), as previously described [13]. For a representative sample, two-dimensional data, ^1^H-^1^H total correlation spectroscopy (TOCSY), and ^1^H-^13^C heteronuclear single quantum coherence (HSQC) were collected for metabolites assignment. 

### 3.3. Metabolomics Data Analysis

Chemical shifts were assigned to metabolites based on 1D ^1^H, 2D TOCSY, and HSQC NMR experiments with reference spectra found in databases, Human Metabolome Database (HMDB) [14] and Chenomx^®^ NMR Suite profiling software (Chenomx Inc., Edmonton, AB, Canada, version 8.4). The concentrations of the metabolites in polar extracts were calculated using Chenomx software based on the internal standard, TMSP. A total of 55 polar metabolites were assigned and quantified. The metabolites (umoles) were then normalized to the tissue weight (g) for statistical analysis. All the multivariate metabolomics analyses were performed using MetaboAnalyst 5.0 [15] using Pareto scaling. The hierarchical clustering analysis was performed using Euclidean distance measuring along with the Ward clustering algorithm to make the heatmap form the top 25 ANOVA values for each gestation. Sparse partial least squares discriminant analysis (sPLS-DA) was then performed on each gestation. Additionally, Q2 performance scores were found for each sPLS-DA model using the LOOCV cross-validation method to understand if these were strongly predictive models.

One-way ANOVA tests were performed on Metaboanalyst for each gestation with post-hoc analysis by both Fisher’s Least Significant Difference method (Fisher’s LSD) and Tukey’s Honestly Significant Difference (Tukey’s HSD), utilizing an FDR value cutoff of 0.05 to note significant metabolites. In addition, volcano plot analysis was performed on each pair-wise comparison (MMC vs. RA, MMC vs. VC, RA vs. VC) for each gestation. This model combines fold change analysis with a *t*-test to extract the most significant metabolites. The cutoff for significant metabolites was set at a fold change (FC) of 1.0 with an FDR *p*-value threshold of 0.05.

## 4. Results

### 4.1. Metabolic Profile at Mid–Late Gestation (E15)

In an analysis by hierarchical clustering dendrogram (Figure 1A) at mid–late gestation (E15), no clear clustering of the samples within their exposure groups was observed, suggesting less metabolic differences between the sample groups. Contrarily, the sPLS-DA scores plot (Figure 1B) showed separation of the MMC group from the RA and VC group on its *x*-axis, component 1, with a small amount of overlap of the 95% confidence intervals, while there was no separation on the *y*-axis, component 2. A more prominent overlap of 95% confidence intervals was observed between the VC and RA groups, which suggests that the two control groups have more similar metabolic profiles. The corresponding loadings plot (Figure 1C) revealed that the elevation of hypoxanthine, inosine, and niacinamide, along with the depletion of tryptophan and succinate, were most responsible for the separation of the MMC group from the RA and VC groups in component 1. One-way ANOVA plot (Figure 1D) showed hypoxanthine (*p* = 0.04) to be significantly altered, which supports the sPLS-DA model. Other notable metabolites found on sPLS-DA analysis were not found to be significantly altered on the ANOVA plot, likely due to the large variation seen within each exposure group. Additional pair-wise comparisons via volcano plot analysis indicated that hypoxanthine (*p* = 0.04, FC = 2.61), inosine (*p* = 0.04, FC = 2.16), and niacinamide (*p* = 0.04, FC = 1.74) were significantly altered in only the MMC vs. VC comparison while having no significant differences for MMC vs. RA and RA vs. VC comparisons.

### 4.2. Metabolic Profile at Term Gestation (E20)

At late gestation, four of the five MMC samples were clustered together on the hierarchical clustering dendrogram (Figure 2A). The fifth sample was unique enough to be classified into its own group, although it was most closely paired with the group that contained the other four MMC samples. On the other side of the dendrogram, four of the five VC samples were classified together, with VC-1 being grouped with two of the RA samples, while the other three RA samples were classified together. This clustering analysis result suggests that the metabolic alterations during the development of myelomeningocele are more pronounced as the defect progresses to late gestation compared to mid-gestation, as was previously reviewed. The sPLS-DA scores plot (Figure 2B) shows the separation of the MMC group from both of the control groups in component 1 (EV 47.4%) and the separation of the VC group from the MMC and RA groups in component 2 (EV 17.9%). In the loadings plot of component 1 (Figure 2C), the decrease in sn-glycero-3-phosphocholine, glutamine, phenylalanine, glutathione, and pantothenate are the five most responsible for MMC separation from the control groups. For sn-glycero-3-phosphocholine, phenylalanine, glutathione, and pantothenate, the RA group has a lower relative value than the VC group. For glutamine, the VC group has a lower value than the RA group. Notably, there was one sample (MMC20_4) that was extremely different from the rest of the MMC samples on the clustering dendrogram and sPLS-DA, which can potentially be attributed to unrecognized variance in severity of the cerebellar herniation. One-way ANOVA plot analysis at late gestation (Figure 2D) showed 19 significant metabolites, including each of the five most significant metabolites seen on the sPLS-DA analysis listed above (Figure 3). In looking specifically at metabolites related directly to energy production, ADP, AMP, and glucose are all labeled significant on the ANOVA plot, with a relative concentration in the MMC group being lower than in the RA and VC groups. GTP was also labeled significant, though it was due to the VC group concentration of this metabolite being significantly higher than both the MMC and RA groups.

In further understanding these variations between groups, pair-wise analysis by volcano plot was utilized. In a comparison of the MMC and VC groups (Table 1), a total of nine metabolites (ADP, AMP, glutamine, glutathione, GTP, pantothenate, phenylalanine, sn-glycero-3-phosphocholine, and succinate) were found to be significantly decreased in the MMC group. These metabolic changes are in agreement with both the sPLS-DA component 1 loadings and ANOVA results. GTP was also found to be significantly decreased in the RA group when compared to the VC group (*p* = 0.03, FC = 0.55), which indicates that the alteration may be a result of retinoic-acid exposure instead of the MMC development.

### 4.3. Metabolic Effect of Retinoic Acid (RA) Exposure

In some of the metabolic alterations, the change may have been caused, at least partially, by retinoic-acid exposure. As seen on the E20 sPLS-DA plot (Figure 2B), the retinoic acid injected control (RA group) has a clear separation from the vehicle control (VC) group. When looking at the loading plot associated with this time point (Appendix A), inosine, acetate, niacinamide, and tryptophan are the metabolites most responsible for this separation. Additionally, when looking at select metabolites (Appendix A), a lower concentration of ADP and GTP was observed in both the MMC and RA groups compared to the vehicle control group, with only mild nonsignificant differences between the MMC and RA groups. Large variances within RA samples were detected in AMP and succinate that span the difference in concentration between the MMC and VC groups. These patterns of alterations suggest that at least part of the variation seen in these specific metabolites between groups is due to retinoic-acid exposure.

## 5. Discussion

Spina bifida, also known as myelomeningocele, is an extensively studied developmental neural tube defect; however, its pathophysiological effect on the cerebellum has not been widely explored. This study provides insight into the metabolic profile of the Chiari II malformation secondary to myelomeningocele on the cerebellum through the use of a teratogen-induced fetal MMC rat model. Through examining the relative metabolic concentrations seen in these sample groups, this study suggests that the progression of myelomeningocele and hindbrain herniation causes oxidative stress in the cerebellum leading to eventual mitochondrial injury and energetic deficits in late gestation at term. These relative metabolic differences that are seen may be the result of the downregulation of steady-state gene expression, upregulation of a stress-related gene to produce a useful protein, or a variety of other responses in the tissue. Further studies of the gene and protein expression would give a further understanding of the mechanistic changes in this process.

At term, the rat cerebellum appears relatively similar to the human fetal cerebellum in the early second trimester, though Chiari Type II malformations are typically not fully developed in the human fetus at this point [5]. This model of myelomeningocele creation using retinoic acid is well-established, with a high load of retinoids being a known teratogenic substance when given maternally in a variety of models [16,17]. Using this model, we were able to explore the metabolic changes of the cerebellum at mid-late and term gestational points, speaking to a stronger understanding of what the pathophysiological profile of this tissue may look similar to when the Chiari Type II malformation eventually does form in a human fetus. In examining the data, it was clear that as the fetus progressed from mid-late to term gestation, alterations seen amongst the observed metabolites led to separations between the samples that developed myelomeningocele, retinoic-acid exposed samples without myelomeningocele, and control non-exposed samples. As the fetus grows and anatomy begins to change, myelomeningocele clearly has a different metabolic picture taking place when compared to fetuses without anatomical abnormality. This was corroborated across various statistical models, albeit not always statistically significant, with the separation becoming more pronounced in late gestation compared to mid–late gestation. A larger sample size would have been useful in making these differences more significant. Myelomeningocele can have notable variation in the level of herniation of the cerebellum, which may explain some of the variances amongst the myelomeningocele samples, given there was one outlier in the group. However, it was not readily accessible to measure the degree of this Chiari Type II-like herniation at the time of cerebellar harvest.

There were multiple metabolites identified as altered in the myelomeningocele samples that play vital roles in energetic and protective pathways within the neural tissue. Of these, inosine and niacinamide were also heavily altered in the retinoic acid-exposed samples. Retinoic-acid exposure can have varying effects on the tissue that must be accounted for, as there is a wide variety of responses in this model to retinoic-acid exposure [5]. Therefore, the variations seen in both inosine and niacinamide within the myelomeningocele samples can be attributed to being more associated with retinoic-acid exposure than the anatomical abnormality. Inosine gives rise to hypoxanthine within the purine salvage pathway [18]. Inosine may have protective effects against oxidative stress and related mechanisms of injury, including hypoxia, glucose deprivation, and age-related effects, preserving the viability of neuronal tissue [19,20,21,22]. Niacinamide, which is a precursor to NAD/NADP, can function as a strong antioxidant in the mitochondria of the brain [23]. Its addition has been shown to prevent oxidative stress and various forms of neuronal damage in other models [24,25]. This molecule has a clear role in energy production, with its relative elevation in the mid-late gestation myelomeningocele samples potentially stemming from a need for further energy production through the oxidative reactions that this metabolite is needed for, along with its use in other non-redox pathways that prevent neuronal damage [26].

The other altered metabolites that were broadly identified in the myelomeningocele model include hypoxanthine, glutathione, succinate, glutamine, phenylalanine, pantothenate, and sn-glycero-3-phosphocholine. The mechanisms associated with these metabolites include oxidative damage, energetic failure, and neurodegeneration. Hypoxanthine was significantly altered at mid-late gestation, but not late gestation; however, a larger sample size may have allowed the elevated trend seen in late gestation myelomeningocele samples to become a significant difference at both time points. This metabolite is intimately involved in the purine salvage pathway, which is utilized for recycling nucleosides and reforming energy molecules. In high concentrations, it also notoriously induces reactive oxygen species in neural tissue leading to endothelial damage, inflammation, and mitochondrial failure [27,28,29]. It should be noted that, while not significantly altered in direct comparisons at late gestation, hypoxanthine had an elevated trend in myelomeningocele samples that may have become significant with a larger sample size, pointing toward a continued trend in its effect on the tissue as a marker of oxidative stress.

Glutathione is known as a major endogenous antioxidant that prevents oxidative stress through redox reactions, with its depletion contributing to cell injury and eventual death [30,31,32]. Its depletion in these sample comparisons could potentially be attributed to a high level of oxidative stress depleting cerebellar reserves of the molecule, with a lack of new production. Glutamate and glycine both play important roles in the production of glutathione [33]. Both of these molecules have trended toward depletion in the late gestation myelomeningocele samples, with glutamate having significance in some models but not consistently across models, while glycine is insignificant. Even with the lack of corresponding complete alterations of the precursor and energy molecules utilized in glutathione’s function as an antioxidant, this depletion of glutathione itself still speaks to a mechanism of oxidative stress taking place that results in significant utilization and degradation of this molecule, resulting in eventual cellular damage. 

Succinate plays a notable role in energy production through its intimate involvement with the TCA cycle at the electron transport chain. Depletion of this molecule, as seen in late gestation myelomeningocele samples, may point to mitochondrial dysfunction [34]. Succinate is formed through the TCA cycle from succinyl-CoA through the use of GDP. A depletion in both succinate and GTP, as seen in this study, may speak to dysfunction in this part of the TCA cycle. Fumarate, which is made in the TCA cycle from succinate, was elevated in the myelomeningocele group compared to the control group at late gestation, though not significant. It was not altered in comparison to the control groups, as the retinoic-acid-exposed non-myelomeningocele group had a wide amount of variation. This low amount of succinate with a coinciding relatively high amount of fumarate would suggest that fumarate is potentially being produced through alternate pathway utilization, such as from the purine nucleotide cycle and degradation of phenylalanine [35]. This cycle produces fumarate from aspartate through interconversion of IMP into AMP and is thought to function in the brain under conditions that induce loss of ATP [36,37].

Glutamine and phenylalanine both play important roles in the production of neurotransmitters in the brain. Glutamine is utilized to make glutamate and GABA [38]. Additionally, glutamine can be utilized in the mitochondria of astrocytes and neurons for energy production via the glutamate–glutamine cycle, where it is utilized for the production of TCA cycle intermediates, glutamine, or glutathione [39]. Therefore, its depletion in myelomeningocele cerebellum samples could have notable impacts on both energy and antioxidant production. Phenylalanine is an essential amino acid that is a precursor to a variety of important physiological molecules, including tyrosine, dopamine, and serotonin [40]. This amino acid is degraded through both glucogenic and ketogenic pathways, therefore having utility in alternate energy production pathways, including the production of ketone bodies intermediates and fumarate [35,41].

Pantothenate specifically is known to be concentrated in myelinated structures, with a deficiency of pantothenate being seen in disorders of demyelination [42,43]. Notably, it is an important component of coenzyme A, which is utilized in a variety of biological processes, including amino acid, carbohydrate, lipid, and protein metabolism. It is also incorporated into Acetyl-CoA and cholesterol, which can be used for the growth of myelin and energy production in Schwann cells and oligodendrocytes [44,45]. Therefore, the depletion of pantothenate seen at late gestation in these myelomeningocele samples may speak to a similar role of this molecule in myelin development and neurodegeneration in this study, with its loss coinciding with damage to neural tissue. Sn-glycero-3-phosphocholine is a phospholipid intermediate of phosphatidylcholine, which plays an important role in myelin composition and remyelination [46,47]. Its depletion, pantothenate, may additionally speak to myelin injury taking place at this point in the disease process.

Energy molecules, such as GTP, AMP, ADP, and ATP, play a variety of important roles in energy production. While none of these were consistently significantly altered across the various statistical models, a larger sample size could lead to a clearer picture of the raw, energetic potential differences seen in myelomeningocele that appears to be taking place through examination of these significantly altered metabolites.

## 6. Conclusions

In summary, these findings provide new insight into the metabolic alterations seen in the cerebellum associated with myelomeningocele using a retinoic-acid-induced Spina bifida rat model. The metabolic changes seen at late gestation were more prominent than at mid-late gestation, pointing to notable progression in the level of tissue injury throughout development by increased compression of the herniated cerebellar tissue in the narrow bone canal. Through examining the specific metabolic alterations at these two-time points, oxidative stress and energetic depletion appear to be intimately involved in causing cellular injury in the cerebellum during these physiologic processes of Spina bifida. These findings can be utilized to better inform further studies of the mechanisms of cellular injury in Spina bifida, leading to the development of better methods of detection and intervention of the disease in utero. 

## Figures and Tables

**Figure 1 metabolites-13-00670-f001:**
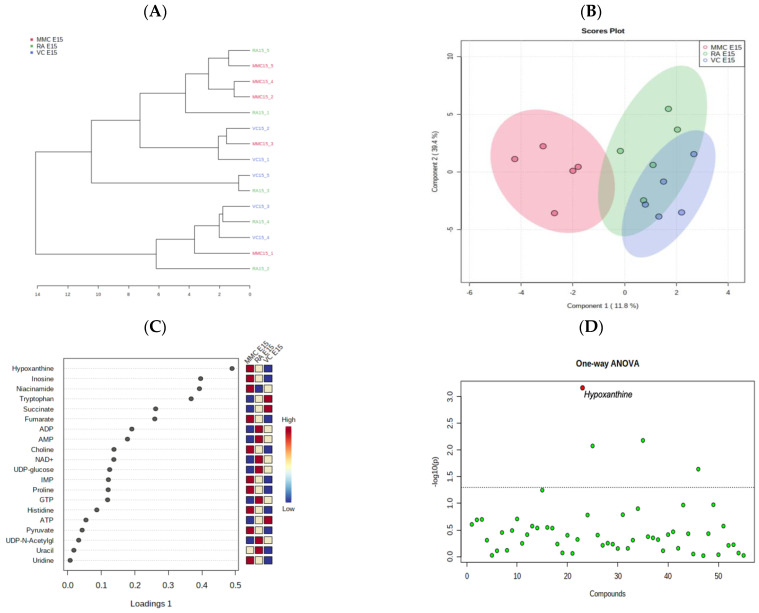
MMC metabolic profile at mid–late gestation (E15). (**A**) Hierarchical Clustering Dendrogram of rat cerebellar metabolome. At mid-gestation, 5 samples were analyzed from Spina bifida (MMC: red), retinoic-acid sham (RA: green), and normal (VC: blue). (**B**) Loading score plot of sparse partial least square discriminant analysis (sPLS-DA), comparing Spina bifida model (MMC: red), retinoic acid sham (RA: green), and normal (VC: blue) with component 1 (EV 11.8), component 2 (EV 39.4%), and 95% confidence intervals with (**C**) the corresponding loadings plot with the variables ranked by absolute values of their loadings. (**D**) One-way ANOVA plot analysis with significant metabolites (*p* < 0.05) labeled red.

**Figure 2 metabolites-13-00670-f002:**
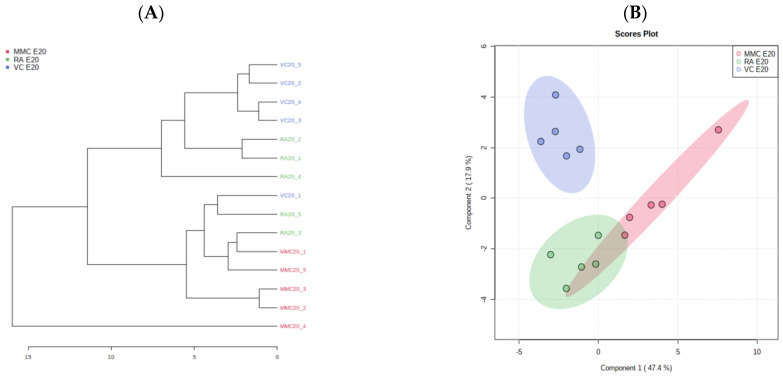
MMC metabolic profile at term gestation (E20). (**A**) Hierarchical Clustering Dendrogram of rat cerebellar metabolome. At term-gestation, 5 samples were analyzed from Spina bifida (MMC: red), retinoic-acid sham (RA: green), and normal (VC: blue). (**B**) Loading score plot of sparse partial least square discriminant analysis (sPLS-DA), comparing Spina bifida model (MMC: red), retinoic acid sham (RA: green), and normal (VC: blue) with component 1 (EV 47.4%), component 2 (EV 17.9%), and 95% confidence intervals with (**C**) the corresponding loadings plot with the variables ranked by absolute values of their loadings. (**D**) One-way ANOVA plot analysis with significant metabolites (*p* < 0.05) labeled red.

**Figure 3 metabolites-13-00670-f003:**
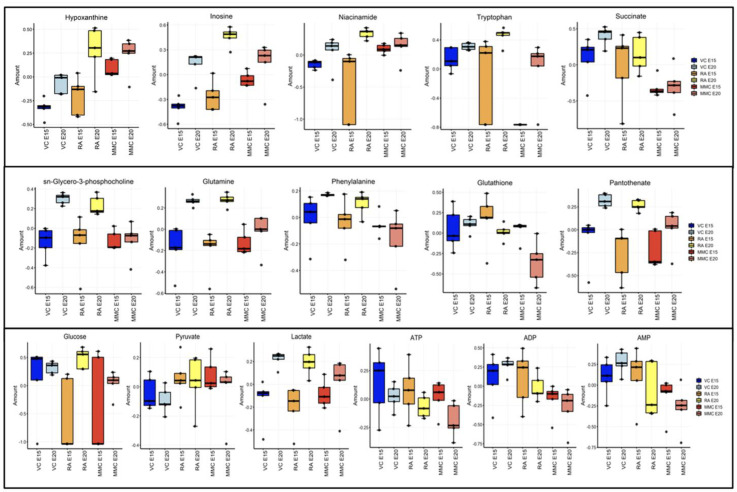
Boxplots of selected metabolites. (**Top row**) Relative concentrations of the top 5 metabolites seen on sPLS-DA analysis of all three groups at mid–late gestation (E15). (**Middle row**) Relative concentrations of the top 5 metabolites seen on sPLS-DA analysis of all three groups at late gestation (E20). (**Bottom**) Relative concentrations of notable metabolites relating to energy production in comparison of all three groups.

**Table 1 metabolites-13-00670-t001:** Significantly altered metabolites on pair-wise comparison of E20 MMC vs. VC. Volcano plot analysis of pair-wise comparison between Spina bifida (MMC) and normal (VC) at late gestation (E20). Nine metabolites were found to be significant (*p* < 0.05, FC > 1.5).

Significantly Altered Metabolites (E20)—MMC vs. VC		
Metabolite	FC	*p* Value (FDR)
ADP	0.31567	0.016892
AMP	0.31439	0.029109
Glutamine	0.53706	0.021256
Glutathione	0.38811	0.021256
GTP	0.43598	0.016892
Pantothenate	0.52425	0.029109
Phenylalanine	0.50395	0.038424
sn-Glycero-3-phosphocholine	0.39806	0.0062079
Succinate	0.22668	0.016892

## Data Availability

The data presented in this study are available on request from the corresponding author. The data are not publicly available due to the author’s preference.

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
