# Peer review of "Non-Targeted Metabolic Profiling of Cerebellum in Spina Bifida Fetal Rats"

_metabolites, 2023, doi:10.3390/metabo13050670_

Round 1

Reviewer 1 Report

In the current study, authors investigated metabolic changes in the cerebellar tissue in a retinoic acid-induced spina bifida rat model by nuclear magnetic resonance (NMR) based metabolomics. They found that the metabolic changes seen at late gestation were more prominent than at mid gestation, pointing to notable progression in level of tissue injury throughout development by increased compression of the herniated cerebellar tissue in the narrow bone canal.

It is an interesting study, but there are several issues that are needed to be considered.

1.It is not clear how this animal model of spinal bifida is established?

2. what was the reason of selecting 2 times of tissue collection? (mid-gestation (labeled E15) and –late-gestation (E20))

3.which statistical test was used for analysis?

4. How many cerebellar tissues were collected per group?

5. LSD post-hoc test is not a suitable test, authors need to try another post hoc such as Tukey or sidak tests.

6. How authors tested normal distribution of data?

7. It is not clear what is the translational aspect of this study? How findings of this study would be helpful in clinic?

no comment

Author Response

Reviewer #1 comments and responses

In the current study, authors investigated metabolic changes in the cerebellar tissue in a retinoic acid-induced spina bifida rat model by nuclear magnetic resonance (NMR) based metabolomics. They found that the metabolic changes seen at late gestation were more prominent than at mid gestation, pointing to notable progression in level of tissue injury throughout development by increased compression of the herniated cerebellar tissue in the narrow bone canal.

It is an interesting study, but there are several issues that are needed to be considered.

1.It is not clear how this animal model of spinal bifida is established?

            Thanks for the comment about the animal model. Spina bifida RA-induced rat model is a well-known and established animal model. As discussed in the introduction on page 2 of the manuscript, the spina bifida rat model has been established in a previous study (Oria 2018, Oria 2023) in our lab. Through gavage of pregnant rats once with retinoic acid 10 days into their gestational period, the previous study induced this malformation in 70% of the fetuses. This paragraph in the introduction has been edited to make this clearer. Additionally, further studies are cited in establishment of the utility of high dose retinoids in development of myelomeningocele across various models.

  1. what was the reason of selecting 2 times of tissue collection? (mid-gestation (labeled E15) and –late-gestation (E20))

            Thanks for the question and this is very important as spina bifida neurodegeneration progresses with gestational age. Two time points were selected to visualize how metabolic differences create divergence of the sample groups as development and myelomeningocele and Chiari type II progresses. Furthermore, E15 in the rat corresponded with 23-25 EGA in humans when spina bifida repair is conducted. That will allow us to understand the metabolic changes before potential treatment.

3.which statistical test was used for analysis?

            Thanks again for the comment related to statistical analysis. All the analysis was performed using a specific software for metabolites analysis (Metaboanalyst 5.0). Software selected Sparse partial least squares discriminant analysis (sPLS-DA) was utilized for statistical modeling with LOOCV cross validation method. Other analytical models that were utilized included one-way ANOVA, Fisher’s LSD post-hoc analysis, and volcano plot analysis.

  1. How many cerebellar tissues were collected per group?

            For each group (myelomeningocele, sham, and control), 5 cerebellar tissues were collected at each time point. This made for a total of 30 tissue samples. Now is clarified and Tissue Collection in point number 3. Procedure. Thanks

  1. LSD post-hoc test is not a suitable test, authors need to try another post hoc such as Tukey or sidak tests.

            Thanks so much for detecting this error as the software provides the correct analysis. Univariate analysis methods are the most common methods used for exploratory data analysis. For multigroup analysis, MetaboAnalyst provides one-way Analysis of Variance (ANOVA). As ANOVA only tells whether the overall comparison is significant or not, it is usually followed by post-hoc analyses in order to identify which two levels are deferent. MetaboAnalyst provides two most commonly used methods for this purpose - Fisher's least significant difference method (Fisher's LSD) and Tukey's Honestly Significant Deference (Tukey's HSD). The univariate analyses provide a preliminary overview about features that are potentially significant in discriminating the conditions under study. The manuscript has been edited to reflect that the software utilizes both of these methods in analyzing the data set.

  1. How authors tested normal distribution of data?

            We totally agree with the reviewer that normalization of the data it’s extremely important for the right quantification of the metabolites in a tissue. That is why the samples were normalized to the tissue weight in grams for statistical analysis and pareto scaling was used normalizing the data on Metaboanalyst 5.0. Procedure described in Metabolomics data analysis section.

  1. It is not clear what is the translational aspect of this study? How findings of this study would be helpful in clinic?

            The translational aspect of this study is that by further studying the mechanisms of cellular injury that were exposed in this study, medical science may be able to better develop methods of detection and intervention that can reduce or eliminate progression of the disease in utero. In addition, E15 in rat gestational age corresponded with the 23-25 weeks of human gestation when currently fetal surgery spina bifida repair is scheduled in clinical practice. The conclusion has been edited to reflect this answer. Furthermore, another reason we characterized E15 and E20 as comparation with human early-mid and late gestation, respectively based on organogenesis development. Lung development is known as an indicator of maturity and organogenesis when comparing humans and rodents. Lung development predominantly occurs during the canalicular-saccular phase in mice from E17 to birth. Comparatively, this reflects changes that occur during weeks 15-38 of gestation in humans (Jackson CM et al. Frontiers in Immunology 2020). Additionally, spinal cord neurogenesis begins at E11 and finishes between E16-E17 with the start of gliogenesis at E17 in rodents (Bayer et al. NeuroToxicology 1993); therefore, these time points made sense in order to study the progression of the defect in response Chiari type II malformation in Spina Bifida.

Reviewer 2 Report

This is a good manuscript.

1. Could the authors please discuss more the correlation between metabolites and gene expression? What is the relevance of the metabolic changes? Is it a reflection of the general preexisting cellular processes like detoxification or protection? Or is it the outcome of the activation of the specific gene set? 

2. Can metabolic profiling describe the mechanisms of the diseases or are these changes unspecific alterations? I am asking because the changed metabolites described here are very similar to almost any metabolomics study.  

3. How many of the rats actually developed spina bifida (SB)? The authors do not provide any evidence about this. They only mention that RA induces SB in 70% of rats. 

4. How were the RA treatments performed? A single dose or were rats treated for many days? The authors should provide a figure showing the treatment schedule. 

5. How good is the used rat strain for the model? Maybe different strains have different susceptibilities for RA? The authors should discuss this part as well.

6. My main concern is that while the title refers to spina bifida, the authors did not present evidence about spine bifida.

7. How the sex effects were adjusted or controlled?

Author Response

Reviewer #2 comments and responses

This is a good manuscript.

  1. Could the authors please discuss more the correlation between metabolites and gene expression? What is the relevance of the metabolic changes? Is it a reflection of the general preexisting cellular processes like detoxification or protection? Or is it the outcome of the activation of the specific gene set?

            Metabolic concentrations are the result that we can observe in response to changes in gene expression. The visualized metabolic variations we see could be the result of a reduction in preexisting enzyme operation or increased production of an enzyme in response to cellular injury. In the case of spina bifida and cerebellar tissue, it is likely a combination of these effects, however further protein analysis is necessary to correlate metabolic changes with specific enzymes and gene regulation. The discussion has been edited to reflect this answer.

  1. Can metabolic profiling describe the mechanisms of the diseases or are these changes unspecific alterations? I am asking because the changed metabolites described here are very similar to almost any metabolomics study.  

            We totally agreed with the reviewer point that metabolic profile it’s just a reflection of the current alteration in the cerebellum in a specific time point. These changes could guide us to understand the potential mechanistic changes during gestation and to recognize in the future if these changes revert with treatment. In the context of these spina bifida samples and the divergence of metabolite concentrations that can be seen between different sample groups from mid to late gestation, mechanisms related to disease can be inferred however further protein and genetic analysis would be necessary in better describing those mechanisms. Thank you for the question and this is reflected in the discussion.

  1. 3. How many of the rats actually developed spina bifida (SB)? The authors do not provide any evidence about this. They only mention that RA induces SB in 70% of rats.

            Thanks for the question about the animal model used in this study. RA-Induced Spina Bifida is a well-known and used animal model in congenital malformations. This animal model is used routinely in our lab and the % incidence was published in a previous work (Oria et al 2018) already cited in the manuscript. The incidence varies depending on each lab, dose, and time of administration and that is why we edited the animal information to clarify these points.

  1. How were the RA treatments performed? A single dose or were rats treated for many days? The authors should provide a figure showing the treatment schedule.

            Thanks for the comment about the animal model. Spina bifida RA-induced rat model is a well-known and established animal model. As discussed in the introduction on page 2 of the manuscript, the spina bifida rat model has been established in a previous studies (Oria 2018, Oria 2023) in our lab. Through gavage of pregnant rats once with retinoic acid 10 days into their gestational period, the previous study induced this malformation in 70% of the fetuses. This paragraph in the introduction has been edited to make this clearer. Additionally, further studies are cited in establishment of the utility of high dose retinoids in development of myelomeningocele across various models. Also, a Supplementary figure is added for better understanding of the model.

  1. How good is the used rat strain for the model? Maybe different strains have different susceptibilities for RA? The authors should discuss this part as well.

            Thank you for this constructive criticism of the model studied. We agree that there are multiple neural tube defects animal models that we could have chosen for this study; however, we ultimately chose this well-established and cost efficient teratogenic induced model of spina bifida to avoid any genetic effect in our analysis. Furthermore, using the retinoic acid induced spina bifida model, we were able to easily obtain siblings exposed to the teratogen that did not develop spina bifida (labeled RA). This internal control group and a vehicle only group allowed us to differentiate the effect of the Chiari type II from the teratogen effect.

  1. My main concern is that while the title refers to spina bifida, the authors did not present evidence about spine bifida.

            Thanks for the concern and we are willing to show more clarity for the reviewers and potential readers of the manuscript. Myelomeningocele and spina bifida are interchangeable names and this entire study was based off analysis of samples with this disease versus control groups. We added images of spina bifida animals and Chiari type II cerebellums in the supplementary Figure 1 which also includes the animal model.

  1. How the sex effects were adjusted or controlled?

            Thanks so much for such important point of discussion but unfortunately, we did not control gender un our analysis.

Round 2

Reviewer 1 Report

Authors responded the comments in a satisfactory manner.